# Enhancing Target-unspecific Tasks through a Features Matrix

**Fangming Cui** [1]  **Yonggang Zhang** [2]  **Xuan Wang** [3]  **Xinmei Tian** [4]  **Jun Yu**[†5]

## Abstract

Recent developments in prompt learning of large Vision-Language Models (VLMs) have significantly improved performance in target-specific tasks. However, these prompting methods often struggle to tackle the target-unspecific or generalizable tasks effectively. It may be attributed to the fact that overfitting training causes the model to forget its general knowledge. The general knowledge has a strong promotion on target-unspecific tasks. To alleviate this issue, we propose a novel Features Matrix (FM) approach designed to enhance these models on target-unspecific tasks. Our method extracts and leverages general knowledge, shaping a Features Matrix (FM). Specifically, the FM captures the semantics of diverse inputs from a deep and fine perspective, preserving essential general knowledge, which mitigates the risk of overfitting. Representative evaluations demonstrate that: 1) the FM is compatible with existing frameworks as a generic and flexible module, and 2) the FM significantly showcases its effectiveness in enhancing target-unspecific tasks (base-to-novel generalization, domain generalization, and cross-dataset generalization), achieving state-of-the-art performance.

## 1. Introduction

Large vision-language models such as CLIP (Radford et al., 2021) have attracted increasing attention for remarkable generalization performance. Vision-Language Models (VLMs) are trained to align textual and visual modalities by leveraging extensive datasets. For instance, a prominent example of such models is CLIP, which has achieved remarkable success across a wide range of downstream tasks (Greer et al., 2024; Sanghi et al., 2022; Etchegaray et al., 2024).

[1]Shanghai Jiao Tong University [2]Hong Kong Baptist University [3]Meituan Inc. [4]University of Science and Technology of China [5]Harbin Institute of Technology (Shenzhen). Correspondence to: Jun Yu[†] <yujun@hit.edu.cn>.

*Proceedings of the $42^{nd}$ International Conference on Machine Learning*, Vancouver, Canada. PMLR 267, 2025. Copyright 2025 by the author(s).

CLIP utilizes a large collection of 400 million text-image pairs. One of CLIP's most appealing features is its ability to perform zero-shot inference. During inference, CLIP utilizes hand-crafted text inputs, known as prompts, to generate classification weights to predict image features, all without requiring any target-specific parameters training.

In contrast to hand-crafted prompts, a model-parameter tuning method has been proposed as prompt learning to automatically learn prompt embeddings (Zhou et al., 2022a). For instance, CoOp (Zhou et al., 2022b) represents the first method that specifically focuses on learning the text embeddings of prompts with few-shot samples training while keeping the CLIP model frozen. Although CoOp (Zhou et al., 2022b) has shown significant performance improvements over hand-crafted prompts in target-specific base classes, it may yield inferior performance compared to the hand-crafted prompt CLIP in generalization tasks, e.g., the novel class of generalization from base-to-novel (see Table 1). To overcome this challenge, some effective methods (Lu et al., 2022; Zhu et al., 2023; Zhou et al., 2022a;b; Yao et al., 2023a; Chen et al.; Li et al., 2024a) with tuning textual embeddings have been proposed. These methods aim to enhance the performance of novel classes, surpassing the novel performance of the previous CoOp. Although the novel ability of these methods surpasses that of CoOp (67.96%), it is unfortunate that the novel ability of these methods still falls short compared to the hand-crafted prompt method (CLIP: 74.22%), as shown in Table 1 (Novel). A possible reason (Zhou et al., 2022a) may be that these methods with tuning text embeddings tend to overfit the downstream data distributions. This overfitting can result in the model losing its inherent generalization capabilities (Yao et al., 2023a) obtained from hand-crafted prompts (Radford et al., 2021).

To alleviate this problem, we propose a novel Features Matrix (FM) with CLIP for target-unspecific tasks. Our method incorporates multiple hand-crafted prompts to extract general information as a pre-trained matrix from frozen CLIP to enhance generalization. The matrix of pre-trained features can delve into the semantics of different hand-crafted prompts finely and deeply, which reduces the risk of forgetting the essential general knowledge of pre-trained CLIP. Importantly, our method is compatible with current prompt learning frameworks for textual or multi-modal prompt learning and serves as a flexible and generic module. As

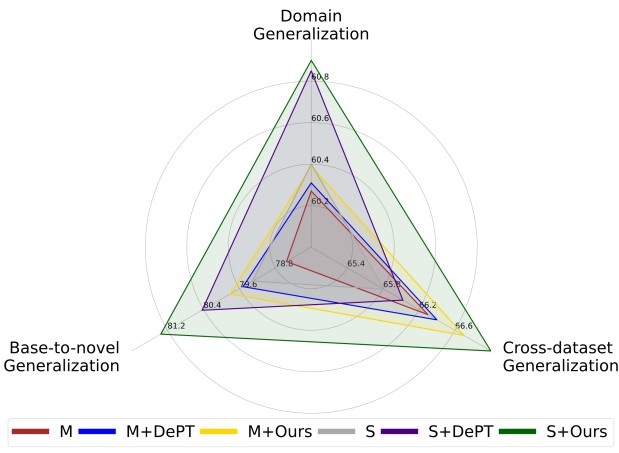

*Figure 1.* Our method is orthogonal to representative architectures, such as MaPLe (M) and PromptSRC (S), surpassing the existing easy-to-use DePT (Zhang et al., 2024a) by a significant margin.

demonstrated in Figure 1, our method, integrated with the representative MaPLe (Khattak et al., 2023a) and Prompt-SRC (Khattak et al., 2023b), exhibits competitive results in base-to-novel generalization, domain generalization, and cross-dataset generalization across 11 datasets. Meanwhile, our method surpasses the existing easy-to-use DePT (Zhang et al., 2024a) by a significant margin. This underscores the robustness of our method across various tasks.

Our contributions can be summarized as follows:

- The proposed method incorporates multiple hand-crafted prompts with classes to extract general knowledge as a pre-trained Features Matrix (FM).

- We propose an easy-to-use design, which is compatible with representative textual or multi-modal prompt learning frameworks for adapting CLIP.

- Various target-unspecific tasks (base-to-novel generalization, cross-dataset generalization, and domain generalization) across 11 datasets demonstrate that our method demonstrates its effectiveness.

## 2. Preliminaries

**Notations.** Considering a pre-trained VLM, let $\mathcal{E}_v(\cdot)$ be its image encoder and $\mathcal{E}_t(\cdot)$ be its text encoder. The image encoder transforms input images into feature embeddings, capturing the visual information within the images. The text encoder generates representations for word embedding sequences, capturing the semantic information conveyed by the text prompts $\mathbf{p}$. Generally, a hand-crafted prompt $\mathbf{p}$ may have the form of "a photo of a [Class]". In this paper, $\mathbf{x}$ represents an arbitrary image, and $l$ denotes the label.

**Hand-Crafted CLIP.** During the pre-training phase of CLIP (Radford et al., 2021), the image and text encoders are jointly trained on large-scale text-image pairs using a contrastive loss. This loss maximizes the cosine similarity between matching pairs and minimizes it between non-matching pairs, enabling the encoders to learn aligned visual and textual representations effectively. The final prediction score between the image $\mathbf{x}$ and text prompt $\mathbf{p}$ is computed using contrastive learning. The final prediction probability of alignment is computed by the matching score as follows:

$$p(l = k \mid \mathbf{x}) = \frac{\exp\left\{\cos\left(\mathcal{E}_t\left(\mathbf{p}_k\right), \mathcal{E}_v(\mathbf{x})\right)/\tau\right\}}{\sum_{k'=1}^{K} \exp\left\{\cos\left(\mathcal{E}_t\left(\mathbf{p}_{k'}\right), \mathcal{E}_v(\mathbf{x})\right)/\tau\right\}}, \quad (1)$$

where $l$ is the label of $\mathbf{x}$, $\cos(\cdot, \cdot)$ stands for cosine similarity between two vectors, and $\tau > 0$ represents a temperature parameter. Here, the classifier consists of $K$ textual features derived from prompts $\{\mathbf{p}_{k'}\}_{k'=1}^{C}$, where the prompt $\mathbf{p}_{k'}$ for the $k'$-th class may have the form of "a photo of a".

**Textual Prompting.** In textual prompting, the class name is retained as prior knowledge, while the word embeddings (referred to as context) of prompts are treated as learnable parameters, as shown in CoOp (Zhou et al., 2022b) and Co-CoOp (Zhou et al., 2022a) of Figure 2. By modeling these context embeddings as trainable parameters, the model can optimize the prompts based on the specific requirements, enhancing the alignment between images and prompts. For class $k$, the tuning feature of the text encoder is denoted as $t_k^{tun}$ in a dataset with a total of $K$ classes. The final prediction probability of cross-entropy loss for two-modalities alignment is computed as:

$$p(l = k \mid \mathbf{x}) = \frac{\exp\left\{\cos\left(t_k^{tun}, \mathcal{E}_v(\mathbf{x})\right)/\tau\right\}}{\sum_{k'=1}^{K} \exp\left\{\cos\left(t_{k'}^{tun}, \mathcal{E}_v(\mathbf{x})\right)/\tau\right\}}, \quad (2)$$

where $\cos(\cdot, \cdot)$ stands for cosine similarity, and $\tau$ denotes a temperature parameter. In some learned text embedding methods, the method aligns with the CoOp methodology by setting the embeddings to be shared across different classes. During the inference phase, the prompts, along with the learned embeddings, can generate textual features that are used for classification purposes. By leveraging these learned embeddings, the model can produce more effective representations, leading to improved target-specific classification performance (Zhou et al., 2022b).

**Multi-Modal Prompting.** Based on textual deep prompting, this design uses only a limited number of trainable parameters based on the image encoder, as shown in MaPLe (Khattak et al., 2023a) and PromptSRC (Khattak et al., 2023b) of Figure 2. The visual prompts are introduced at every transformer layer's input space. In the visual branch, the input image $\mathbf{x} \in \mathbb{R}^{C \times H \times W}$ is divided into $M$ patches. And, a tuning embedding of class $i_{class}$ is appended with the input

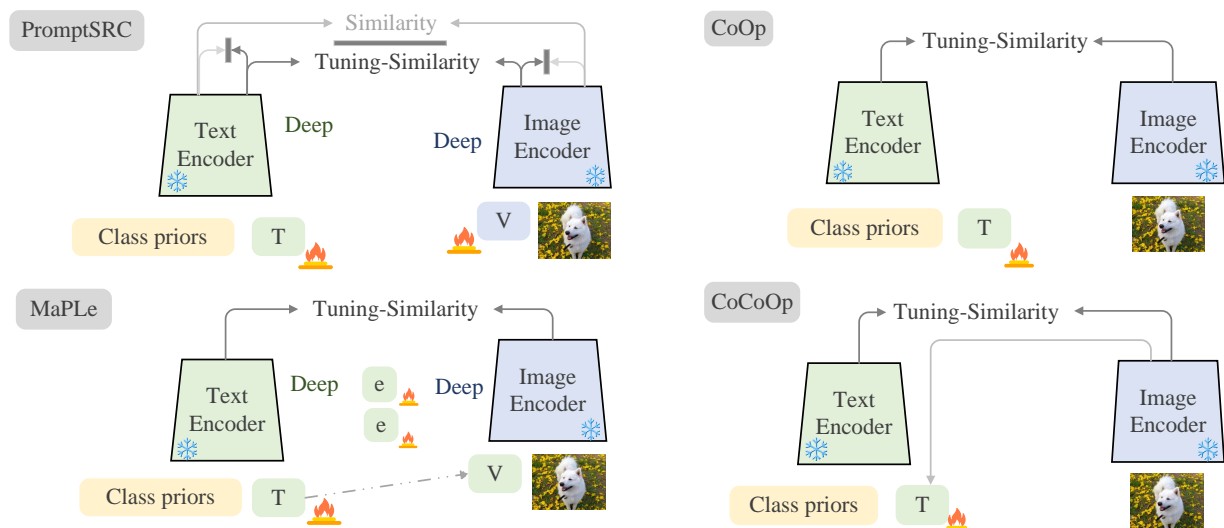

*Figure 2.* Illustration of representative textual prompting frameworks (CoOp and CoCoOp) and multi-modal prompting frameworks (MaPLe and PromptSRC). We propose a flexible and generic design, which is compatible with these representative architectures. In the figure, "snowflake pattern" represents parameter freezing, "flame pattern" represents learnable pattern, "Deep" represents learnable tokens embedded in several layers of the encoder, where "T" represents learnable text embedding, "V" represents learnable visual embedding, "Class priors" represents which category it belongs to, "Tuning Similarity" represents the calculation of cosine similarity for fine-tuning architecture, and the light gray "Similarity" represents the calculation of cosine similarity for frozen architecture. In the MaPLe architecture diagram, the "e" represents the matrix function that connects the encoders of two modalities in several layers.

patches. The foundation of vision prompting is ViT (Vision Transformer) (Dosovitskiy et al., 2020) backbone, which shares the same image encoder as CLIP. Let $\mathcal{P}_v$ denote the visual embeddings of prompts, and the image encoder processes the input tokens to generate tuning visual features, as follows (Jia et al., 2022):

$$\tilde{\mathbf{x}}_p = \{\mathcal{P}_v, \boldsymbol{i}_{class}, \boldsymbol{i}_1, \boldsymbol{i}_2, \cdots, \boldsymbol{i}_M\}. \qquad (3)$$

The tuning of visual embeddings are introduced in the image encoder for deep prompt tuning.

## 3. Methodology

### 3.1. Current Challenge

As shown in Table 1, such as textual prompting (Zhou et al., 2022a;b; Yao et al., 2023a; Lu et al., 2022) and multi-modal prompting (Chen et al.), have shown significant improvements in target-specific base classes. However, these methods often suffer from overfitting, leading to poor generalization in novel classes. To improve the novel performance of models, prompting regularization involves constraining the pre-trained and fine-tuning features in the text branch through computed loss. This constraint helps prevent forgetting general knowledge and ensures that the model retains essential information for downstream tasks. The objective of this design is to activate and maintain the pre-trained general knowledge, leveraging its remarkable generalization

abilities to mitigate performance degradation when dealing with target-unspecific classes.

*Table 1.* Our method is an easy-to-use design, integrated into CoOp, obtaining a higher average performance (11 datasets) on novel (target-unspecific) classes.

| Method | Hand Features | Base | Novel |
|---|---|---|---|
| CLIP (ICML2021) | Single | 69.34 | **74.22** |
| CoCoOp (CVPR2022) | No | 80.47 | 71.69 |
| ProDA (CVPR2022) | No | 81.56 | 72.30 |
| PLOT (ICLR2023) | No | 81.24 | 72.98 |
| ProGrad (ICCV2023) | No | 82.48 | 70.75 |
| CoOp (IJCV2022) | No | 82.38 | 67.96 |
| + DePT (CVPR2024) | No | 83.66 | 71.82 |
| + Kg (CVPR2023) | Single | 80.73 | 72.70 |
| + Ours | Matrix | 81.15 | **74.66** |

It is important to recognize that the fixed CLIP's fundamental pre-trained characteristics demonstrate robust generalization capabilities. Recently, KgCoOp (Yao et al., 2023a) incorporates the constraint with $L1$ loss on fine-tuning feature with the pre-trained feature for the textual branch, avoiding the loss of CLIP's original generalization capability.

**Motivation.** However, the novel class performance of handcrafted-based KgCoOp (72.70%) is still lower than pretrained CLIP (74.22%) (Radford et al., 2021) of Table 1. It

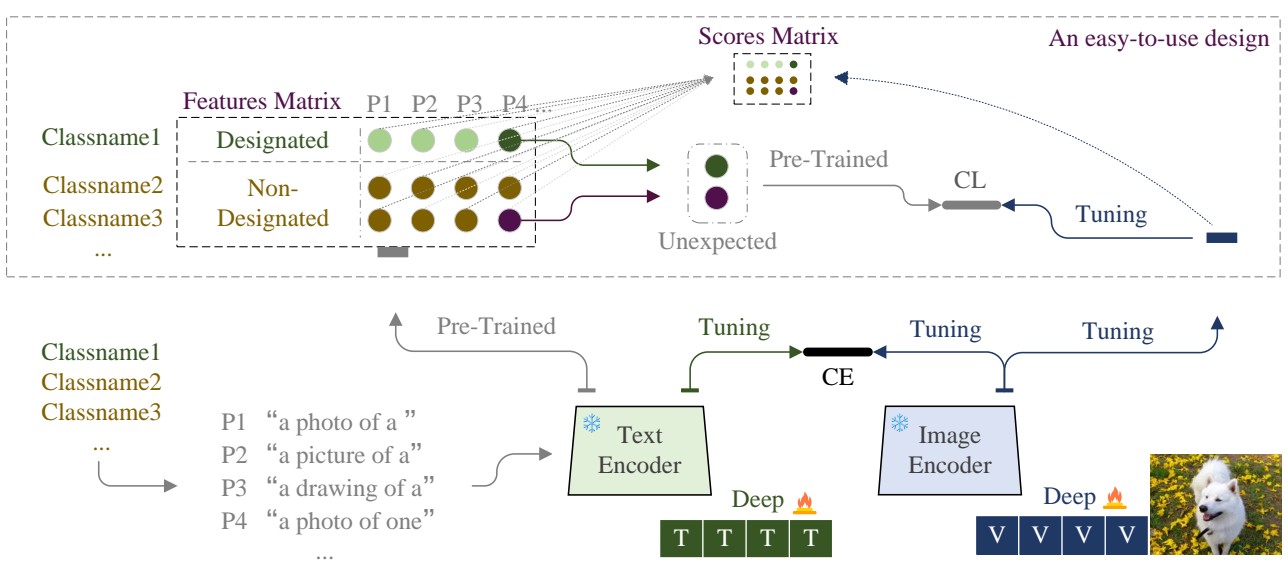

*Figure 3.* Illustration of our easy-to-use method. We propose a novel Features Matrix (FM) for enhancing target-unspecific tasks. Our method incorporates multiple hand-crafted prompts with classes to extract general knowledge as a pre-trained features matrix. Various generalization tasks across 11 datasets demonstrate that our method outperforms existing prompt learning methods.

may be attributed to the fact that this single hand-crafted prompting regularization with CLIP cannot fully release and excavate the diverse semantics for utilizing essential general knowledge (Khattak et al., 2023b).

### 3.2. Proposed Features Matrix (FM)

To tackle this challenge and problem, we propose a novel design called Features Matrix (FM) for enhancing target-unspecific tasks. Our method incorporates multiple hand-crafted prompts with classes of the same datasets to extract general information as a pre-trained features matrix from CLIP to enhance generalization. In this way, we can explore and excavate the semantic information brought by each hand-crafted prompt more finely and deeply. By specifically aligning these multiple pre-trained hand-crafted unexpected features and tuning image features, our method aims to enhance the model's generalization and robustness. This involves focusing on indistinguishable features that pose a challenge to the model during training. As shown in Figure 3, we employ a set of hand-crafted prompt templates (P1, P2, P3, P4, etc) as inputs in the text encoder of pre-trained frozen CLIP. These prompt templates, such as "a photo of one", "a photo of a ", "a picture of a", "a drawing of a", etc. The number of prompt templates (Radford et al., 2021) is set to 60. In a dataset containing $K$ classes, the input to the text encoder is the frozen embeddings of hand-crafted prompts, consisting of different classes. The output of the frozen text encoder can be viewed as a features matrix, as depicted in Figure 3. Specifically, we let $t$ denote the sets of pre-trained text features extracted by a frozen text encoder

through a set of hand-crafted prompt templates (P1, P2, P3, P4, etc) with classes. In a dataset with a total of $K$ classes, features belonging to the current label $k$ of image samples are referred to as designated features $t_k$, while features from other classes are considered non-designated features $t_{\hat{k}}$.

Our method computes the matching scores with the cosine distance for each visual output feature $v^{tun}$ and designated pre-trained text features of the features matrix as $\cos(t_k, v^{tun})$. Similarly, we compute the matching scores between each visual output feature and the non-designated pre-trained text features, denoted as $\cos(t_{\hat{k}}, v^{tun})$. In correspondence with the features matrix, these matching scores can be viewed as a scores matrix, as depicted in Figure 3. Subsequently, we identify the textual feature with lower scores (low-$\beta$) among the designated features, which is referred to as the unexpected designated features set $F_{un}^k$. The low-$\beta$ is that there is a list of scores sorted from low to high, with $\beta$ values assigned from the front of the list. Similarly, we identify textual features with higher scores (top-$\beta$) among the non-designated features, which are referred to as unexpected non-designated feature sets $F_{un}^{\hat{k}}$. Accordingly, the contrastive loss $\mathcal{L}_{CL}$ of unexpected features similarity is denoted as follows,

$$-\log \frac{\exp\{\cos(t_k, v^{tun})\}}{\exp\{\cos(t_k, v^{tun})\} + \exp\{\cos(t_{\hat{k}}, v^{tun})\}}, \quad (4)$$

where the $t_k \in F_{un}^k$ and $t_{\hat{k}} \in F_{un}^{\hat{k}}$ denote the selected unexpected features for designated unexpected features and non-designated unexpected features. The objective is to explore and excavate the unexpected features of general

information. We train the objective for aligning the un-expected text features and tuning visual features, and the contrastive loss $\mathcal{L}_{\text{CL}}$ optimizes visual embeddings. For two-modal alignment, let $\mathcal{L}_{\text{CE}}(\cdot, \cdot)$ denote the cross-entropy loss with prompts $\mathbf{p}$ for samples $\mathcal{S}$, as follows,

$$\mathcal{L}_{\text{CE}} = \arg\min_{\mathbf{p}} \mathbb{E}_{(\mathbf{x},y)\sim\mathcal{S}} \mathcal{L}\left\{\cos\left(t^{tun}, v^{tun}\right), l\right\}. \quad (5)$$

Accordingly, the objective $\mathcal{L}_{\text{total}}$ with a hyper-parameter $\gamma$ of our method can be formulated as follows,

$$\mathcal{L}_{\text{total}} = \mathcal{L}_{\text{CE}} + \gamma\mathcal{L}_{\text{CL}}. \quad (6)$$

Consequently, optimizing models with $\mathcal{L}_{\text{CL}}$ between the pre-trained textual unexpected features and image tuning features. The $\mathcal{L}_{\text{CE}}$ represents two modalities of alignment of deep vision-language prompting. Importantly, our method is compatible with representative prompt learning frameworks as a generic and flexible module for textual prompt learning, such as CoOp and CoCoOp, or multi-modal prompt learning, such as MaPLe and PromptSRC.

# 4. Main Generalization Tasks

We evaluate our method on generalization tasks. As evidenced by systematic benchmarking in Figure 1, our framework, when synergistically integrated with MaPLe (Khattak et al., 2023a) and PromptSRC (Khattak et al., 2023b), achieves state-of-the-art performance across three critical generalization dimensions (base-to-novel generalization, domain generalization, and cross-dataset generalization) spanning 11 heterogeneous benchmarks. Notably, our method surpasses the existing easy-to-use DePT (Zhang et al., 2024a) by a significant margin in generalization scenarios through low-resource training.

## 4.1. Benchmark Setting

**Compared Methods.** In three generalization experiments, we compare our method with CoOp (IJCV2022) (Zhou et al., 2022b), CoCoOp (CVPR2022) (Zhou et al., 2022a), MaPLe (CVPR2023) (Khattak et al., 2023a), Prompt-SRC (ICCV2023) (Khattak et al., 2023b), and DePT (CVPR2024) (Zhang et al., 2024a). The DePT is an easy-to-use method. We process frozen CLIP using a linear probe method. Comparison methods are based on the ViT-B/16 architecture for fair comparison.

**Implementation Details.** We employ CLIP (Radford et al., 2021) model based on the ViT-B/16 architecture. For the PromptSRC-based and MaPLe-based frameworks, we set the visual and textual embedding length to 4. We set the easy-to-use module $\gamma$ to 0.1 and matching scores (top and low) $\beta$ to 5. Training for 30 epochs for a base-to-novel setting in the first 9 transformer layers. Training for 20

epochs for the domain generalization setting and the cross-dataset evaluation setting in the first 3 transformer layers. We use a SGD optimizer with a learning rate of 0.0025 on a single GPU. Following the hand-crafted CLIP, we use the hand-crafted template sets from (Radford et al., 2021).

**Datasets.** In Table 10, the datasets cover multiple recognition tasks including ImageNet (Deng et al., 2009) and Caltech101 (Fei-Fei et al., 2004) which consists of generic objects, OxfordPets (Parkhi et al., 2012), StanfordCars (Krause et al., 2013), Flowers102 (Nilsback & Zisserman, 2008), Food101 (Bossard et al., 2014), and FGVCAircraft (Maji et al., 2013) for fine-grained classification, SUN397 (Xiao et al., 2010) for scene recognition, UCF101 (Soomro et al., 2012) for action recognition, DTD (Cimpoi et al., 2014) for texture classification, and EuroSAT (Helber et al., 2019) which consists of satellite images. We leverage datasets such as ImageNetA (Hendrycks et al., 2021b), ImageNet-R (Hendrycks et al., 2021a), ImageNet-Sketch (Wang et al., 2019), and ImageNetV2 (Recht et al., 2019) to assess the model's performance across different domain distributions.

## 4.2. Base-to-Novel Generalization Task

In the base-to-novel generalization task, the datasets are divided into base and novel classes. The model is trained on the base classes in a 16-shot setting, and tested on both the base and novel classes across 11 different datasets. The number of classes for base and novel is the same, which means that all classes in a dataset are evenly divided into two groups of classes. The process of dividing all classes in the dataset is randomly selected. HM refers to harmonic mean. The HM evaluates the generalizable and non-generalizable ability of our methods.

**Discussion.** In Table 2, our method demonstrates significant improvements on all 11 datasets on HM. We found that our performance is higher than DePT (Zhang et al., 2024a) (CVPR2024) when integrated with different methods. Compared to DePT, our method significantly enhances the performance of base classes while also improving the accuracy on novel classes. In terms of average performance, our method based on PromptSRC (S) attains 85.70% accuracy on the base classes, and 77.35% accuracy on the novel classes. Our method, as an easy-to-use module, achieves performance improvements for various baseline frameworks, such as CoOp, CoCoOp, MaPLe (M), and PromptSRC (S).

## 4.3. Domain Generalization Task

We train our model using the ImageNet in 16 shots and we leverage ImageNetA, ImageNet-R, ImageNet-Sketch, and ImageNetV2 to assess the model's performance. The training ImageNet is a non-generalizable setting in Table 3 (column-1). This experiment aims to validate the potential of our method in domain shifts (Fang et al., 2023).

*Table 2.* Base-to-novel generalization. In the base-to-novel generalization task, the datasets are divided into base and novel classes. The model is trained on the base classes in a 16-shot setting, and tested on both the base and novel classes across 11 different datasets. Compared with easy-to-use DePT (CVPR2024) in detail, our method achieves consistent average performance improvement over different representative baselines (CoOp, CoCoOp, MaPLe, PromptSRC).

(a) **Average over 11 datasets**

|        | Base  | Novel | HM    |
|--------|-------|-------|-------|
| CoOp   | 82.69 | 63.22 | 71.66 |
| +DePT  | **83.66** | 71.82 | 77.29 |
| + Ours | 81.15 | **74.66** | **77.79** |
| Co     | 80.47 | 71.69 | 75.83 |
| +DePT  | **83.80** | 72.89 | 77.97 |
| + Ours | 81.68 | **75.55** | **78.52** |
| M      | 82.28 | 75.14 | 78.55 |
| +DePT  | **84.85** | 74.82 | 79.52 |
| + Ours | 84.45 | **76.53** | **80.32** |
| S      | 84.26 | 76.10 | 79.97 |
| + DePT | 85.19 | 76.17 | 80.43 |
| + Ours | **85.70** | **77.35** | **81.32** |

(b) ImageNet

|        | Base  | Novel | HM    |
|--------|-------|-------|-------|
| CoOp   | 76.47 | 67.88 | 71.92 |
| +DePT  | 77.13 | 70.10 | 73.45 |
| + Ours | 75.85 | 71.33 | 73.53 |
| Co     | 75.98 | 70.43 | 73.10 |
| +DePT  | 76.87 | 70.47 | 73.53 |
| + Ours | 77.35 | 72.36 | 74.79 |
| M      | 76.66 | 70.54 | 73.47 |
| +DePT  | 77.87 | 70.23 | 73.85 |
| + Ours | 78.18 | 71.38 | 74.62 |
| S      | 77.60 | 70.73 | 74.01 |
| + DePT | 78.20 | 70.27 | 74.02 |
| + Ours | **78.90** | **71.58** | **75.07** |

(c) Caltech101

|        | Base  | Novel | HM    |
|--------|-------|-------|-------|
| CoOp   | 98.00 | 89.81 | 93.73 |
| +DePT  | 98.33 | 94.33 | 96.29 |
| + Ours | 97.58 | 96.60 | 97.13 |
| Co     | 97.96 | 93.81 | 95.84 |
| +DePT  | 98.37 | 93.87 | 96.06 |
| + Ours | 98.61 | 96.75 | 97.70 |
| M      | 97.74 | 94.36 | 96.02 |
| +DePT  | 98.53 | 95.03 | 96.75 |
| + Ours | 98.35 | 96.11 | 97.22 |
| S      | 98.10 | 94.03 | 96.02 |
| + DePT | 98.57 | 94.10 | 96.28 |
| + Ours | **98.62** | **95.88** | **97.27** |

(d) OxfordPets

|        | Base  | Novel | HM    |
|--------|-------|-------|-------|
| CoOp   | 93.67 | 95.29 | 94.47 |
| +DePT  | 94.70 | 97.63 | 96.14 |
| + Ours | 93.78 | 97.80 | 95.78 |
| Co     | 95.20 | 97.69 | 96.43 |
| +DePT  | 94.03 | 97.20 | 95.59 |
| + Ours | 95.33 | 98.15 | 96.76 |
| M      | 95.43 | 97.76 | 96.58 |
| +DePT  | 95.03 | 97.83 | 96.41 |
| + Ours | 95.85 | 98.22 | 97.04 |
| S      | 95.33 | 97.30 | 96.30 |
| + DePT | 95.43 | 97.33 | 96.37 |
| + Ours | **95.95** | **97.92** | **96.95** |

(e) EuroSAT

|        | Base  | Novel | HM    |
|--------|-------|-------|-------|
| CoOp   | 92.19 | 54.74 | 68.69 |
| +DePT  | 88.27 | 66.27 | 75.70 |
| + Ours | 88.35 | 65.33 | 75.13 |
| Co     | 87.49 | 60.04 | 71.21 |
| +DePT  | 90.27 | 66.87 | 76.82 |
| + Ours | 88.15 | 70.11 | 78.12 |
| M      | 94.07 | 73.23 | 82.35 |
| +DePT  | 94.43 | 76.23 | 84.36 |
| + Ours | 94.22 | 75.65 | 83.93 |
| S      | 92.90 | 73.90 | 82.32 |
| + DePT | 92.23 | **77.90** | 84.88 |
| + Ours | **95.50** | 76.85 | **85.17** |

(f) UCF101

|        | Base  | Novel | HM    |
|--------|-------|-------|-------|
| CoOp   | 84.69 | 56.05 | 67.46 |
| + DePT | 85.43 | 72.17 | 78.24 |
| + Ours | 83.10 | 78.85 | 80.94 |
| Co     | 82.33 | 73.45 | 77.64 |
| + DePT | 85.70 | 72.80 | 78.73 |
| + Ours | 82.95 | 77.21 | 80.00 |
| M      | 83.00 | 78.66 | 80.77 |
| + DePT | 86.87 | 78.10 | 82.25 |
| + Ours | 87.33 | 79.10 | 83.02 |
| S      | 87.10 | 78.80 | 82.74 |
| + DePT | 87.73 | 77.70 | 82.46 |
| + Ours | **88.80** | **79.50** | **83.93** |

(g) StanfordCars

|        | Base  | Novel | HM    |
|--------|-------|-------|-------|
| CoOp   | 78.12 | 60.40 | 68.13 |
| + DePT | 79.67 | 72.40 | 75.86 |
| + Ours | 74.32 | 76.87 | 75.58 |
| Co     | 70.49 | 73.59 | 72.01 |
| + DePT | 79.87 | 73.33 | 76.46 |
| + Ours | 72.88 | 76.10 | 74.46 |
| M      | 72.94 | 74.00 | 73.47 |
| + DePT | 80.93 | 71.73 | 76.06 |
| + Ours | 78.66 | 75.13 | 76.86 |
| S      | 78.27 | 74.97 | 76.58 |
| + DePT | 80.80 | 75.00 | 77.79 |
| + Ours | **80.91** | **76.51** | **78.68** |

(h) Flowers102

|        | Base  | Novel | HM    |
|--------|-------|-------|-------|
| CoOp   | 97.60 | 59.67 | 74.06 |
| + DePT | 98.20 | 72.00 | 83.08 |
| + Ours | 96.22 | 72.32 | 82.57 |
| Co     | 94.87 | 71.75 | 81.71 |
| + DePT | 98.33 | 72.57 | 83.51 |
| + Ours | 95.61 | 74.93 | 84.03 |
| M      | 95.92 | 72.46 | 82.56 |
| + DePT | 98.03 | 73.17 | 83.79 |
| + Ours | 98.21 | 75.00 | 85.07 |
| S      | 98.07 | 76.50 | 85.95 |
| + DePT | 98.40 | 77.10 | 86.46 |
| + Ours | **98.81** | **78.10** | **87.26** |

(i) Food101

|        | Base  | Novel | HM    |
|--------|-------|-------|-------|
| CoOp   | 88.33 | 82.26 | 85.19 |
| + DePT | 90.43 | 91.33 | 90.88 |
| + Ours | 89.98 | 92.85 | 91.41 |
| Co     | 90.70 | 91.29 | 90.99 |
| + DePT | 90.30 | 91.30 | 90.80 |
| + Ours | 90.61 | 91.93 | 91.28 |
| M      | 90.71 | 92.05 | 91.38 |
| + DePT | 90.33 | 91.53 | 90.93 |
| + Ours | 90.31 | 92.81 | 91.57 |
| S      | 90.67 | 91.53 | 91.10 |
| + DePT | **90.87** | 91.57 | 91.22 |
| + Ours | 90.61 | **92.30** | **91.45** |

(j) FGVCAircraft

|        | Base  | Novel | HM    |
|--------|-------|-------|-------|
| CoOp   | 40.44 | 22.30 | 28.75 |
| + DePT | 42.53 | 22.53 | 29.46 |
| + Ours | 37.32 | 34.61 | 35.92 |
| Co     | 33.41 | 23.71 | 27.74 |
| + DePT | 43.07 | 31.30 | 36.25 |
| + Ours | 37.87 | 34.91 | 36.33 |
| M      | 37.44 | 35.61 | 36.50 |
| + DePT | 44.53 | 32.80 | 37.78 |
| + Ours | 42.46 | 37.62 | 39.89 |
| S      | 42.73 | 37.87 | 40.15 |
| + DePT | 45.70 | 36.73 | 40.73 |
| + Ours | **45.81** | **39.11** | **42.20** |

(k) SUN397

|        | Base  | Novel | HM    |
|--------|-------|-------|-------|
| CoOp   | 80.60 | 65.89 | 72.51 |
| + DePT | 82.37 | 75.07 | 78.55 |
| + Ours | 79.12 | 78.38 | 78.77 |
| Co     | 79.74 | 76.86 | 78.27 |
| + DePT | 82.20 | 76.73 | 79.37 |
| + Ours | 80.32 | 79.00 | 79.68 |
| M      | 80.82 | 78.70 | 79.75 |
| + DePT | 82.90 | 76.40 | 79.52 |
| + Ours | 82.35 | 79.81 | 81.07 |
| S      | 82.67 | 78.47 | 80.52 |
| + DePT | 83.27 | 78.97 | 81.06 |
| + Ours | **83.90** | **80.51** | **82.20** |

(l) DTD

|        | Base  | Novel | HM    |
|--------|-------|-------|-------|
| CoOp   | 79.44 | 41.18 | 54.24 |
| + DePT | 83.20 | 56.13 | 67.04 |
| + Ours | 77.10 | 56.38 | 65.15 |
| Co     | 77.01 | 56.00 | 64.85 |
| + DePT | 82.77 | 55.40 | 66.37 |
| + Ours | 78.90 | 59.61 | 67.93 |
| M      | 80.36 | 59.18 | 68.16 |
| + DePT | 83.87 | 59.93 | 69.91 |
| + Ours | 83.01 | 60.98 | 70.32 |
| S      | 83.37 | 62.97 | 71.75 |
| + DePT | 84.80 | 61.20 | 71.09 |
| + Ours | **84.90** | **62.58** | **72.07** |

*Table 3.* Domain generalization. We train our model using the ImageNet in 16 shots and test its performance on 4 different variants of the ImageNet. When compared to DePT, our method consistently outperforms it across all ImageNet variant datasets.

| | Source | Target | | | | |
|---|---|---|---|---|---|---|
| | ImageNet | -V2 | -S | -A | -R | Average |
| CoOp | 71.51 | 64.2 | 47.99 | 49.71 | 75.21 | 59.28 |
| + DePT | 72.63 | 64.80 | 48.05 | 50.00 | 75.50 | 59.58 |
| + Ours | 71.82 | 65.13 | 48.10 | 50.15 | 76.13 | **59.87** |
| CoCoOp | 71.02 | 64.07 | 48.75 | 50.63 | 76.18 | 59.91 |
| + DePT | 72.77 | 65.10 | 49.10 | 51.00 | 76.85 | 60.51 |
| + Ours | 72.10 | 65.25 | 50.13 | 52.11 | 77.18 | **61.16** |
| MaPLe | 70.72 | 64.07 | 49.15 | 50.9 | 76.98 | 60.27 |
| + DePT | 73.27 | 65.33 | 49.05 | 51.25 | 77.50 | 60.78 |
| + Ours | 71.56 | 65.45 | 50.33 | 52.32 | 78.10 | **61.55** |
| PromptSRC | 71.27 | 64.35 | 49.55 | 50.90 | 77.80 | 60.65 |
| + DePT | 71.60 | 64.51 | 50.15 | 51.88 | 77.18 | 60.93 |
| + Ours | 71.41 | **65.50** | **51.33** | **52.00** | **78.85** | **61.92** |

**Discussion.** In Table 3, our method based on PromptSRC demonstrates the highest average improvement of 1.27% over the PromptSRC method. Additionally, when compared to DePT, our method consistently outperforms it across all ImageNet variant datasets.

### 4.4. Cross-Dataset Generalization Task

We train our model with 16 shots on the ImageNet dataset and test the model on 10 other unseen datasets. The training ImageNet is a non-generalizable setting in Table 4 (column-1). This experiment aims to validate the potential of our method in a wide range of cross-dataset transfers.

**Discussion.** As shown in Table 4, our method based on PromptSRC shows competitive performance in 10/10 over the generic and flexible method DePT. These findings suggest that our method excels in achieving better generalization across a diverse range of unseen datasets. However, our method for training ImageNet is lower than that of DePT.

## 5. Further Studies

In this section, we discuss further studies, focusing on the impact of embedding length and depth, computational cost, applying to other ViT instances, analysis of $\gamma$ for $\mathcal{L}_{CL}$, and analysis of Top-$\beta$ and Low-$\beta$ for pre-trained features matrix.

### 5.1. Learning Depth

In Table 5, we note that increasing the learning depth generally increases the performance based on PromptSRC. As the number of layers increases to 11, the HM value decreases.

### 5.2. Embeddings Length

In Table 7, our findings indicate that the performance based on PromptSRC reaches its peak when the length of embeddings is set to 4 on HM for an average of 11 datasets. Our ablation studies are based on keeping all other settings unchanged. It indicates excessive fine-tuning of the model, causing it to lose CLIP generalization.

### 5.3. Analysis of $\gamma$ for $\mathcal{L}_{CL}$

In Table 9, it is observed that as the number of $\gamma$ increases to 0.1, there is a peak in HM for an average of 11 datasets. We conduct analysis based on the PromptSRC model.

### 5.4. Analysis of Top-$\beta$ and Low-$\beta$

In Table 8, it is observed that as the number of Top-$\beta$ and Low-$\beta$ increases to 5, there is a peak in HM. We conduct analysis based on the PromptSRC model for an average of 11 datasets. We keep other hyper-parameters unchanged.

### 5.5. Applying to Other ViT Instances

In Table 6, we apply our method in various types of ViT instances. Our experimental performance is based on the average performance of 11 datasets. Our generic and flexible module based on PromptSRC has the lowest performance of ViT-B/32, with a value of 79.70%. And, our method, which leverages PromptSRC, achieves the best performance with ViT-L/14, with a value of 84.20%.

### 5.6. Inference Stage Computational Cost

In Table 11, the compute cost analysis is performed using the SUN397 dataset over 10 epochs on a single GPU. Our method (row 3) may have a slower inference speed as a result of multiple cosine similarity calculations. The number of parameters we introduced has remained relatively consistent and stable in training. This implies that usually, our parameter quantity is lower than that of DePT.

## 6. Related Work

**Vision-Language Models (VLMs).** Recently, researchers have demonstrated strong generalization capability of Vision-Language Models (VLMs) (Alayrac et al., 2022; Shao et al., 2025), which involve training on large-scale datasets of image-text pairs. Using a large number of samples for training is indeed one of the most effective methods (Yang et al., 2021; Wang et al., 2023). Such as CLIP (Radford et al., 2021), which is a prominent and straightforward framework among existing VLMs. The strong generalization capability of CLIP has made it a foundation for many methods in adapting pre-trained VLMs for downstream tasks (Sanghi et al., 2022; Maaz et al., 2022;

*Table 4.* Cross-dataset generalization. Our method achieves overall favorable performance.

| | Source | Target | | | | | | | | | | |
|---|---|---|---|---|---|---|---|---|---|---|---|---|
| | ImageNet | Caltech101 | OxfordPets | StanfordCars | Flowers102 | Food101 | Aircraft | SUN397 | DTD | EuroSAT | UCF101 | Average |
| CoOp (Zhou et al., 2022b) | 71.51 | 93.70 | 89.14 | 64.51 | 68.71 | 85.30 | 18.47 | 64.15 | 41.92 | 46.39 | 66.55 | 63.88 |
| + DePT | 72.63 | 93.30 | 90.00 | 65.53 | 70.50 | 85.97 | 21.90 | 66.07 | 43.17 | 44.97 | 68.80 | 65.02 |
| + Ours | 71.82 | 94.10 | 90.33 | 65.82 | 70.01 | 86.10 | 20.71 | 65.11 | 43.98 | 46.55 | 68.00 | **65.07** |
| CoCoOp (Zhou et al., 2022a) | 71.02 | 94.43 | 90.14 | 65.32 | 71.88 | 86.06 | 22.94 | 67.36 | 45.73 | 45.37 | 68.21 | 65.74 |
| + DePT | 72.77 | 94.10 | 90.63 | 66.23 | 72.17 | 86.27 | 22.90 | 67.30 | 45.50 | 44.17 | 69.53 | 65.88 |
| + Ours | 72.10 | 94.88 | 90.57 | 65.80 | 72.15 | 87.00 | 22.78 | 68.12 | 45.98 | 46.15 | 69.10 | **66.25** |
| MaPLe (Khattak et al., 2023a) | 70.72 | 93.53 | 90.49 | 65.57 | 72.23 | 86.20 | 24.74 | 67.01 | 46.49 | 48.06 | 68.69 | 66.30 |
| + DePT | 73.27 | 92.53 | 90.10 | 64.60 | 70.10 | 85.57 | 23.63 | 66.40 | 45.03 | 40.13 | 67.53 | 64.56 |
| + Ours | 71.56 | 94.00 | 90.91 | 65.92 | 73.10 | 86.78 | 24.33 | 68.35 | 46.13 | 49.01 | 68.81 | **66.73** |
| PromptSRC (Khattak et al., 2023b) | 71.27 | 93.60 | 90.25 | 65.70 | 70.25 | 86.15 | 23.90 | 67.10 | 46.87 | 45.50 | 68.75 | 65.81 |
| + DePT | 71.60 | 93.80 | 90.13 | 66.00 | 70.93 | 86.27 | 24.30 | 67.23 | 46.60 | 45.83 | 69.10 | 66.02 |
| + Ours | 71.41 | **94.95** | **92.14** | **66.50** | **73.25** | **87.53** | **25.81** | **68.10** | **49.01** | **49.20** | **69.74** | **67.62** |

*Table 5.* Analysis of learning depth based on PromptSRC for an average of 11 datasets. HM refers to harmonic mean.

| Learning Depth | 1 | 3 | 5 | 7 | **9** | 11 |
|---|---|---|---|---|---|---|
| HM | | 77.11 | 78.03 | 79.12 | 80.01 | **81.32** | 80.21 |

*Table 7.* Analysis of embeddings length based on PromptSRC for an average of 11 datasets. HM refers to harmonic mean.

| Prompt Length | 1 | 2 | **4** | 6 | 8 | 10 |
|---|---|---|---|---|---|---|
| HM | | 77.01 | 80.05 | **81.32** | 80.30 | 79.11 | 78.54 |

*Table 6.* Applying to other ViT instances.

| | Avg. (11 datasets) | | |
|---|---|---|---|
| | B/32 | B/16 | L/14 |
| HM (ViT) | 79.70 | 81.32 | **84.20** |

*Table 8.* Identifying scores of Top-$\beta$ and Low-$\beta$ for an average of 11 datasets. We conduct analysis based on the PromptSRC (Khattak et al., 2023b) model. HM refers to harmonic mean.

| | Avg. (11 datasets) | | | |
|---|---|---|---|---|
| | 3 | 4 | **5** | 6 |
| HM ($\beta$) | 78.21 | 79.81 | **81.32** | 80.51 |

*Table 9.* Analysis of $\gamma$ for $\mathcal{L}_{CL}$ in pre-trained features matrix. We conduct analysis based on the PromptSRC (Khattak et al., 2023b) model for an average of 11 datasets. HM refers to harmonic mean.

| | Avg. (11 datasets) | | | |
|---|---|---|---|---|
| | 0.05 | **0.1** | 0.5 | 0.9 |
| HM ($\gamma$) | 80.13 | **81.32** | 81.00 | 79.88 |

Bangalath et al., 2022; Zhang et al., 2021; Wang et al., 2022; Xu et al., 2025). To enhance the generalization ability of VLMs, researchers have explored various approaches (Zhou et al., 2025b;a). Some approaches involve enhancing the text encoder or the visual encoder (Vaswani et al., 2017). By improving the capabilities of the text encoder, the model can better capture the semantics and contextual information in the textual input (Zhang et al., 2024b). Similarly, enhancing the visual encoder allows the model to extract discriminative features from the visual input.

**Prompt Tuning in VLMs.** Prompt tuning (Zhang & Tian, 2025; Pan et al., 2024; Cao et al., 2025; Zhou et al., 2024; Li et al., 2024a) is a commonly employed technique in the field of Natural Language Processing (NLP) for training on downstream tasks (Yu et al., 2025; 2024b;a). Leveraging text prompts, which are instructions given to the language model component of VLMs, is a prevalent practice to improve

task comprehension (Cui et al., 2025b). Full fine-tuning and linear probes are two commonly employed approaches for adapting VLMs to downstream tasks (Fang et al., 2025; Liu et al., 2025). The constraints of both methods have prompted research into innovative techniques influenced by prompt tuning within the realm of VLMs (Yin et al., 2025). CoOp (Zhou et al., 2022b), ProDA (Lu et al., 2022), and

*Table 10.* Training and testing datasets. The dataset consists of 11 image classification datasets and four variant datasets of ImageNet.

| Dataset | Classes | Train | Val | Test |
|---|---|---|---|---|
| ImageNet (Deng et al., 2009) | 1,000 | 1.28 M | N/A | 50,000 |
| Caltech101 (Fei-Fei et al., 2004) | 100 | 4,128 | 1,649 | 2,465 |
| OxfordPets (Parkhi et al., 2012) | 37 | 2,944 | 736 | 3,669 |
| StanfordCars (Krause et al., 2013) | 196 | 6,509 | 1,635 | 8,041 |
| Flowers102 (Nilsback & Zisserman, 2008) | 102 | 4,093 | 1,633 | 2,463 |
| Food101 (Bossard et al., 2014) | 101 | 50,500 | 20,200 | 30,300 |
| FGVCAircraft (Maji et al., 2013) | 100 | 3,334 | 3,333 | 3,333 |
| SUN397 (Xiao et al., 2010) | 397 | 15,880 | 3,970 | 19,850 |
| DTD (Cimpoi et al., 2014) | 47 | 2,820 | 1,128 | 1,692 |
| EuroSAT (Helber et al., 2019) | 10 | 13,500 | 5,400 | 8,100 |
| UCF101 (Soomro et al., 2012) | 101 | 7,639 | 1,898 | 3,783 |
| **-V2** (Recht et al., 2019) | 1,000 | N/A | N/A | 10,000 |
| **-Sketch** (Wang et al., 2019) | 1,000 | N/A | N/A | 50,889 |
| **-A** (Hendrycks et al., 2021b) | 200 | N/A | N/A | 7,500 |
| **-R** (Hendrycks et al., 2021a) | 200 | N/A | N/A | 30,000 |

*Table 11.* The compute cost analysis is performed using the SUN397 (Xiao et al., 2010) dataset over 10 epochs on a single GPU. 'N': the num of classes in the base task (Zhang et al., 2024a).

| Method | Train time | Learnable para. | HM |
|---|---|---|---|
| CoOp | 10.88min | 8K | 71.65 |
| + DePT | 10.91min | + (2+N/2)K | 77.30 |
| + Ours | 13.56min | + 0K | **78.66** |

CoCoOp (Zhou et al., 2022a) fine-tune the CLIP model for few-shot image recognition by optimizing a continuous set of embeddings within the textual branch. The image-conditional prompt utilized in CoCoOp significantly contributes to improving generalization to unseen classes (Peng et al., 2025; 2023). By conditioning prompts on visual features, CoCoOp (Zhou et al., 2022a) ensures that the language model focuses on pertinent visual information when making predictions. Moreover, some approaches (Yao et al., 2023b;a; Cui et al., 2024; 2025a) constrain the learnable prompts to contain the essential general knowledge. In addition, the approach (Zhang et al., 2024a) introduces a flexible approach to align its vision and language representations.

## 7. Future Works and Limitations

In the future, we plan to investigate the potential of our method in other tasks (Tian et al., 2024; Yang et al., 2023; Li et al., 2019; Yu et al., 2023; Li et al., 2024c; Zhang et al., 2022; Wang et al., 2021; Li et al., 2025a;c; Jia et al., 2025; Liu et al., 2023b;a; Li et al., 2024b) and downstream scenarios (Pan et al., 2025; Jin et al., 2025; Li et al., 2025b; Liu et al., 2024; Wu et al., 2024; Liu et al., 2023c;d). Due to multiple processes of our method, the training speed is slower. And, the ImageNet training in Table 4 is lower.

## 8. Conclusion

Prompt learning is a promising method for adapting pre-trained Visual-Language Models (VLMs) for target-specific classification tasks. However, these optimization frameworks exhibit limited efficacy when applied to target-unspecific or generalizable scenarios. This performance degradation may stem from overfitting-induced catastrophic forgetting, where the model loses generalizable knowledge critical for adapting to unseen tasks during the fine-tuning process. In this paper, we propose a Features Matrix (FM) for vision-language models. Our method incorporates multiple hand-crafted prompts to extract general information as a pre-trained features matrix from CLIP to enhance generalization. The matrix of pre-trained features can delve into the semantics of different hand-crafted prompts at a more profound level, which reduces the risk of forgetting the essential general knowledge of pre-trained CLIP. Our method focuses on the challenge of various target-unspecific tasks and scenarios across a wide range of real datasets. Importantly, our method is compatible with representative prompt learning frameworks for textual or multi-modal prompts as a generic and flexible module.

## Acknowledgement

XM T was supported in part by the NSFC No. 62222117. J Y was supported in part by the NSFC No. 62125201 and U24B20174.

## Impact Statement

The societal implications of our work involve democratizing the availability of potent AI resources, as our plug-and-play method can achieve impressive performance even in the absence of extensive labeled data.

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
