# OpenReview forum: "Enhancing Target-unspecific Tasks through a Features Matrix"
_ICML.cc/2025/Conference — ICML 2025 poster_

### Official Review · Reviewer_gDtW · 2025-03-12

**Overall Recommendation:** 3

**Summary:**

Partial parameter optimization methods face challenges in handling target-unspecific tasks due to overfitting, which causes the model to lose its general knowledge essential for these tasks. To address this issue, this paper proposes a regularization technique using a Feature Matrix (FM). Through extensive evaluations across various tasks, the proposed method demonstrates significant improvements in enhancing performance on target-unspecific tasks.

**Claims And Evidence:**

The paper asserts that partial parameter optimization struggles with novel classes due to overfitting, as evidenced by the results in Table 1. To address this issue, the proposed Feature Matrix (FM) enhances generalization by preserving general knowledge through multiple hand-crafted prompts. Experimental results demonstrate that FM significantly improves novel class accuracy compared to previous approaches.

**Essential References Not Discussed:**

I am not a strict expert in this domain, so I cannot fully confirm whether the paper includes all the necessary references. However, it appears to cite the relevant works and uses them effectively for comparison to validate its proposed method.

**Experimental Designs Or Analyses:**

The overall experimental setup appears to be well-designed, and the paper includes essential analyses, such as base and novel classification performance, the impact of additional parameters, prompt length, and cost comparisons with previous methods.

**Methods And Evaluation Criteria:**

The proposed methods are straightforward and easy to comprehend. The paper evaluates their effectiveness using a diverse set of benchmark datasets.

**Other Comments Or Suggestions:**

No other suggestions.

**Other Strengths And Weaknesses:**

- The paper introduces the Feature Matrix (FM) regularization technique to enhance the generalization of unseen classes in prompt learning. The proposed methods are well-structured, intuitive, and easy to understand.
- To validate their approach, the paper evaluates performance across various benchmarks and includes essential analyses, such as hyperparameter tuning and cost comparisons.
- However, a notable limitation is the lack of theoretical analysis, which could strengthen the explanation of why FM improves generalization.

**Questions For Authors:**

Overall, the work is well-structured, easy to understand, and presents a plausible approach. The proposed methods effectively enhance performance on novel classes compared to previous approaches. One notable limitation is the lack of theoretical analysis, which could provide a stronger justification for why the proposed methods improve generalization. However, this does not seem to be a critical issue in evaluating the paper. Since I am not an expert in this domain, I believe it would be best to wait for comments from other reviewers for a more comprehensive assessment.

**Relation To Broader Scientific Literature:**

This paper builds on previous research in vision-language models (VLMs), prompt tuning, and generalization techniques. It also introduces the Feature Matrix (FM) as a novel regularization method to enhance generalization in target-unspecific tasks.

**Theoretical Claims:**

It appears that the paper does not present theoretical claims that require formal proof.

---

> ### Author Rebuttal · Authors · 2025-03-27
>
> # **Thanks for review**:
>  Thank you very much for the reviewer's appreciation of our work. Thank you to the reviewer for your valuable time for our paper, we express our **heartfelt** gratitude.
>
> ## Response：
>
> Our work is based on pre-trained CLIP (ICML2021), and learnable and non learnable prompt words to explore and attempt under existing limited conditions and theories.
>
> * Our initial idea is to **minimize** the introduction of new theories, and to explore performance based on existing technology.
>
> * We focused more on exploring the performance of fine-tuning multimodal models on large-scale datasets for **realistic** application scenarios (11 datasets).
>
> * Our method is a plugin based approach, which appears to have practical value for **convenience** and low coupling.
>
> Thank you to the reviewer for the insights on the theoretical aspects of our work. The theoretical analysis  will definitely increase the strictness of our work. According to the **guidance** of the reviewer, we will continue to improve this manuscript, including but not limited to theoretical analysis of  experiments.
>
> We **promise** to  make revisions before submitting the final version.

---

### Official Review · Reviewer_HJ59 · 2025-03-12

**Overall Recommendation:** 3

**Summary:**

The paper proposes to mitigate the overfitting observed in parametric optimization methods when optimizing on the target domain. To address this issue, the authors introduce a feature matrix-based approach. This method leverages features extracted from multiple handcrafted prompts, combined with features from various classes, and employs a contrastive loss to optimize the process. The aim is to excavate general feature information, although some of it may not be directly relevant to the task. They validate their approach by integrating it with existing methods and evaluating it on multiple datasets. The experimental results, along with the average performance improvements across datasets, demonstrate that their method not only achieves better overall performance and also delivers significant improvements in certain cases.

## update after rebuttal
Thanks for the efforts. I recommend the authors include the full version of Table 8 for every dataset in the paper (as promised). Moreover, the implementational details of the proposed method in conjunction with the other methods are not clear; hence, also include this in the paper from the rebuttal. In response, I have decided to retain my score.

**Claims And Evidence:**

My main critique is about detailing their method (proposal of the paper) and substantiating it with appropriate text and experiments.

 – For the claim that common prompt learning methods suffer from overfitting, leading to poor generalization in novel classes (Lines 162–164), there are no references or empirical evidence. Can the authors provide them to substantiate this?

 – What does "Low β" mean in Line 188? The authors should elaborate.

 – In the methodology, the claim in the abstract regarding the use of regularization is not clear. This term appears in the introduction and abstract but nowhere else in the paper. Can the authors clarify this?

 – The methodology does not explain convincingly why cosine similarity was chosen.

**Essential References Not Discussed:**

No.

**Experimental Designs Or Analyses:**

– While the additional experiments exploring target k-shots and learning depth are interesting, they are insufficient. The authors should include more experiments that provide insights into their method. For instance, demonstrate how many P_{1,2,3} samples are necessary for their method. Similarly, how does the method perform with increases or decreases in t_{k}? What are the assumptions for the prompts, and are there any guidelines?

– I recommend providing the full version of Table 8 that will detail the utility of \beta for every dataset. Currently, with average performance it's unclear which similar features between source and target would be useful

 – The authors vaguely mention the need for a ‘set’ of prompts  (line 377) but fail to specify how many are needed. Can this be clarified and elaborated?

**Methods And Evaluation Criteria:**

Yes.

**Other Comments Or Suggestions:**

Figure 2 doesn't add value to the paper. I recommend shifting it to the appendix.

**Other Strengths And Weaknesses:**

– Can the authors include the algorithm for their method? Additionally, for one of the baselines used, can they incorporate their algorithm into the paper’s algorithm (‘Ours’)? This would provide additional details and help describe the steps more clearly.

 – In the methodology, after Equation 5 is introduced, what happens next? How are the features used in the model for the baseline methods to integrate the proposal? I recommend adding a subsection in the methodology to explain this process clearly. Currently, it is unclear.

**Questions For Authors:**

The authors do not provide enough details regarding methodology and experiments that would investigate their proposal even deeper. As a reader, I was intrigued to know more about features matrix and regularization, which are not detailed sufficiently.

**Relation To Broader Scientific Literature:**

The paper lacks significant contribution in terms of novelty.

**Theoretical Claims:**

The paper includes minimal theory and theory investigation.

---

> ### Author Rebuttal · Authors · 2025-03-27
>
> # **Thanks for review**:
> We express gratitude to the reviewers: 1) for providing rich and **comprehensive** comments of our work, 2) for investing a significant amount of  time and effort in reviewing our work, and 3) for providing practical guidances. We will add acknowledgements in the final version and express our **gratitude**.
>
> ## Response:
>
> >**Q1**: For the claim that common prompt learning methods suffer from overfitting, leading to poor generalization in novel classes (Lines 162–164), there are no references or empirical evidence.
>
>
> **A1**: Thank you very much for the reviewer's reminder. We apologize that our writing here is not very organized. In line (162-164), these methods refer to line (158-160) of paper. These methods are CoCoOp, CoOp, KgCoOp, ProDA, DPLQ, PLOT in line (158-160) of our paper.
>
>
> >**Q2**:  What does "Low-β" mean in Line 188?
>
>
> **A2**: 'Low-β' refers to a list of scores sorted from low to high, with β values assigned from the front. We **promise** to write it clearly in the final version.
>
>
> >**Q3**:  This term 'regularization' appears in the introduction and abstract but nowhere else in the paper.
>
>
> **A3**: Our method is **non-traditional** regularization . In other words, the "regularization" method mentioned in our paper is actually a new method. Based on this, this term 'regularization' does not appear elsewhere in our paper.
>
>
> >**Q4**:  The methodology does not explain convincingly why cosine similarity was chosen.
>
>
> **A4**: Our infrastructure (CLIP) and previous works (CoOp, CoCoOp, ProDA, KgCoOp, PLOT, MaPLe, PromptSRC, etc.) use cosine similarity in the alignment calculation of text and visual branches. Therefore, to ensure **fair** comparison, our method also uses cosine similarity here. Based on this, cosine similarity is also used to align the two modalities in the 'scores matrix' stage. Our idea is to **minimize** the introduction of other measurement methods, and to explore performance based on existing methods (CLIP, CoOp, MaPLe, etc) and hand-crafted templates.
>
>
> >**Q5**:  Demonstrate how many prompt templates are necessary for their method. How does the method perform with increases or decreases in features of templates?
>
>
> **A5**: We used 60 manual prompt templates, the contents of which are in our Appendix (A.6), the number 60 is fixed. We apologize for not including the number in the method details. We **promise** to write it clearly in the final version. Thank you for the careful review by the reviewer.
>
> >**Q6**:  I recommend providing the full version of Table 8 that will detail the utility of $\beta$ for every dataset. Currently, with average performance it's unclear which similar features between source and target would be useful.
>
> **A6**: Thank you for the reviewer's comments. Due to the limited word count of Rebuttal, we now write the specific values of base and novel here to illustrate the relationship between source and target. And, we **promise** to refine the values of each dataset in the final version.
>
> Base-to-novel generalization of 11 datasets ($\beta$)
>  |        $\beta$   |  3      |  4        |  5           | 6           |
>  |:--------------- |:------:|:-------:|:---------:|:----------:|
>  | Base           |  80.19 |  83.47 | **85.70**| 84.46  |
>  | Novel          |  76.33 |  76.51 | **77.35**| 76.90  |
>  | HM             |  78.21 |  79.81 | **81.32**| 80.51  |
>
>
> >**Q7**:  How are the features used in the model for the baseline methods to integrate the proposal? I recommend adding a subsection in the methodology to explain this process clearly.
>
> **A7**: Thank you for the reviewer's comments. In the final version, we promise to add pseudocode for the algorithm flow. In addition, we **promise** to add detailed implementation subsection for plug-and-play applying for CoOp CoOoOp、MaPLe、PromptSRC, and write it into a new subsection. Our brief introduction to the plug-and-play concept is as follows:
>
> * In Figure 3, Equation 5 refers to the part below the grey dashed box. After Equation 5, our plug-and-play framework begins to empower the results with Equation 5. Our plug-and-play architecture input consists of two parts: 1) text features generated by 60 manually prompted words, and 2) visual features generated by learnable visual embeddings. Therefore, learnable visual embeddings are necessary for the work we propose.
>
> * In Figure 2, we found that CoOp and CoCoOp do not have a visual embedding part, so we will first add learnable visual embeddings to the CoOp and CoCoOp architecture. Afterwards, based on this modified architecture, integrate the methods we proposed. For MaPLe and PromptSRC, our proposed method can be directly integrated.
>
> * It is worth noting that the input of the text encoder consists of two parts: 1) manual prompt word templates, and 2) learnable prompt words, which vectorize the text into a learnable matrix. These two parts will not affect each other during the input phase, as they are two separate input processes.

---

### Official Review · Reviewer_5hGa · 2025-03-12

**Overall Recommendation:** 3

**Summary:**

## Summary

This paper addresses out-of-distribution generalization problem in prompt fine-tuning "CLIP" kind of model.  The challenge is common: fine-tuning on a specific task boosts the performance in this task, but hurt the performance on other general tasks.

This work addresses this problem by adding a regularization during fine-tuning. This regularization, working in a constrastive process, encourages the model (during optimization) to output a close representation as the pretrained model.  This is an common, yet effective regularization.

As a regularization, it is supposed to help different "fine-tuning" methods. The experiment shows verifies this hypothesis on many tasks.


-------

## Strengthness
- "negative interference" has long been a problem in model-based algorithm [1]. Fine-tuning in neural network is the common scenario that suffers from "negative interference". Thus it is interesting and important to research on negative interference in fine-tuning neural networks.

- The regularization, which encourages the representation to be close to the pretraining one during fine-tuning, is an resonable and intuitive approach to reduce negative interference.

- Many experimental results justifies the positive effect of the proposed method in this paper.

----

## Weakness

- The novelty of the regularization is questionable. Many related works shares the same idea. For example, L2 weights decay towards pretrained weights rather than zero.  KL-divergence regularization in fine-tuning transformers via reinforcement learning manner, weights-averaging of pretrained and fine-tuned models, etc.




[1] Atkeson, Christopher G., Andrew W. Moore, and Stefan Schaal. "Locally weighted learning." Lazy learning (1997): 11-73.

**Claims And Evidence:**

check summary.

**Essential References Not Discussed:**

no.

**Experimental Designs Or Analyses:**

yes.

**Methods And Evaluation Criteria:**

check summary.

**Other Comments Or Suggestions:**

check summary.

**Other Strengths And Weaknesses:**

check summary.

**Questions For Authors:**

check summary.

**Relation To Broader Scientific Literature:**

check summary.

**Theoretical Claims:**

no theory.

---

> ### Author Rebuttal · Authors · 2025-03-27
>
> # **Thanks for review**:
> Thank you to the reviewer for taking time to take detailed research and valuable reference on our work. We would like to express our **gratitude**. We will also **carefully** revise according to the opinions of the reviewers.
>
> In addition, thank you to the reviewer for providing such a comprehensive reference [1] in the comments, and we went to read this article. This article investigates many restrictive and regularization schemes, which we believe are very valuable. It is interesting and important to research on negative interference. The "negative interference" has long been a problem in model-based algorithm. Our future work will conduct an in-depth investigation of these regularization functions with prompting works. We will add this article to our citation and provide a descriptive introduction to the relevant work.
>
> [1] Atkeson, Christopher G., Andrew W. Moore, and Stefan Schaal. "Locally weighted learning." Lazy learning (1997): 11-73
>
>
> ## Response:
>
> >**Q1**:  L2 weights decay towards pretrained weights rather than zero. KL-divergence regularization in fine-tuning transformers via reinforcement learning manner, weights-averaging of pretrained and fine-tuned models, etc. Many related works shares the idea.
>
>
> **A1**: Thank you very much to the reviewer for specific technical analysis and insights. We will now introduce some **unique** aspects of our work, hoping that reviewer will have new perspectives on our work.
>
> * The **contribution** of our method: fully utilizing manual prompt words to mine information.
>
>
> * Our core module is **non-traditional** sense of regularization. In other words, the "regularization" mentioned in our paper is actually a **new** method.
>
> * Compared to L2 and KL divergence, our method is flexible and **delves** into specific semantics. Our proposed scheme is closely integrated with manual prompt words.
>
> In our future work, we will study and **draw on** the opinions provided by reviewers to design more works in earnest.

---

### Official Review · Reviewer_xBkC · 2025-03-13

**Overall Recommendation:** 3

**Summary:**

This paper proposes a Features Matrix regularization method to improve model performance on target-unspecific tasks. FM preserves general knowledge and reduces overfitting by extracting and leveraging semantic information from diverse inputs. The approach, compatible with existing frameworks, enhances generalization and performs well across various datasets, including 11 datasets with limited labeled data. It also incorporates pre-trained CLIP features and multiple handcrafted prompts to prevent forgetting essential knowledge, demonstrating state-of-the-art performance in target-unspecific tasks.

**Claims And Evidence:**

Yes.

**Essential References Not Discussed:**

No.

**Experimental Designs Or Analyses:**

Yes, I have checked the experimental designs and analyses.

**Methods And Evaluation Criteria:**

Yes.

**Other Comments Or Suggestions:**

Some parts of the paper can be further improved to make the paper more clear, for example, the more detailed discussion of the base classes and novel classes in the introduction. The illustration of different types of prompt design (Fig 2) are not very clear, more details can be added in the figure caption.

**Other Strengths And Weaknesses:**

1. The illustrations are of great style, but the detailed explanation of the illustrations can be improved. The paper writing can be improved to make the paper more clear.
2. The paper discusses the limitation of the proposed method, it is good.
3. The experiments are comprehensive, but more ablation studies are needed to verify the effectiveness of the proposed method.

**Questions For Authors:**

1. What is the details of the split of base classes and novel classes in the experiments?
2. Some ablation studies can be provided to verify the effectiveness of trainable text tokens and image tokens. Because one simple baseline may be to train a "MOE" to dynamically select the manually designed prompt.

**Relation To Broader Scientific Literature:**

This paper proposes a better way to combine manually designed prompts and trainable tokens to improve the performance of target-unspecific tasks.

**Theoretical Claims:**

No, this paper does not have the theoretical claims.

---

> ### Author Rebuttal · Authors · 2025-03-27
>
> # **Thanks for review**:
> We thank the reviewer for the valuable time and consideration of our manuscript. The comments provided by the reviewers are very useful for our work, we express our heartfelt **gratitude**.
>
> ## Response:
>
> >**Q1**:  The detailed explanation of the illustrations can be improved.
>
> **A1**: Thank you very much to the reviewer for important comments on the illustration. Our initial focus in designing this illustration was on "simplicity", "contrast", and "the designed artistic drawing". However, we lack description of some components in the diagram. We **promise** to make serious revisions. Our explanation of the symbols in the figure is as follows:
>
> In the figure, "snowflake pattern" represents parameter freezing, "flame pattern" represents learnable pattern, "Deep" represents learnable tokens embedded in several layers of the encoder, where "T" represents learnable text embedding, "V" represents learnable visual embedding, "Class priors" represents which category it belongs to, "Tuning Similarity" represents the calculation of cosine similarity for fine-tuning architecture, and the light gray "Similarity" represents the calculation of cosine similarity for frozen architecture. In the MaPLe architecture diagram, the "e" represents the matrix function that connects the encoders of two modalities in several layers.
>
> >**Q2**:  The more detailed discussion of the base classes and novel classes in the introduction.
>
> **A2**: We will add the revised records to the final version to express our **gratitude** to the valuable review. In terms of article writing, we need to improve our manuscript from various aspects. We **promise** to make serious revisions. Our explanation of the "base and novel classes" is as follows:
>
> In the base-to-novel generalization task, the datasets are divided into base and novel classes. The model is trained on the base classes, and tested on both the base and novel classes. The number of classes for base and novel is the **same**, which means that all classes in a dataset are evenly divided into two groups of classes. The process of dividing all classes in the dataset is randomly selected.
>
> >**Q3**: Some ablation studies can be provided to verify the effectiveness of trainable text tokens and image tokens.
>
> **A3**: The experiments proposed by the reviewer are very meaningful and have greatly **helped** our work. We will include this part of the experiments in the main paper and express our gratitude to reviewer. We are now conducting ablation experiments on text embedding and visual embedding **separately** to explore more phenomena.
>
> 1) We **add experiments** on the text and image tokens **length** for their effectiveness. Specifically, we set different learning tokens lengths for text tokens and visual tokens. When we conducted ablation experiments on visual embeddings, the value of text embeddings remained at best value (4). The reverse is also the same.
>
> * Base-to-novel generalization of 11 datasets (Textual Tokens Length)
> | Textual Tokens Length     |  1     |  2     |  4       | 6      |  8     | 10    |
> |:---------------             |:------:|:------:|:--------:|:------:|:------:|:------:|
> | HM (Ours based on MaPLe)    |  77.11 |  79.00 | **80.32**| 79.12   | 78.81  | 77.03 |
> | HM (Ours based on PromptSRC) |  78.02 |  79.34 | **81.32**| 81.00   | 80.86  | 78.94 |
>
>
> * Base-to-novel generalization of 11 datasets (Visual Tokens Length)
> | Visual Tokens Length |  1     |  2      |  4       | 6      |  8     | 10    |
> |:---------------       |:------:|:------:|:--------:|:------:|:------:|:------:|
> | HM (Ours based on MaPLe) |  77.01 |  78.11 |**80.32** | 79.33   | 78.11  | 77.50 |
> | HM (Ours based on PromptSRC) |  77.32 |  78.08 |**81.32** | 80.54   | 80.51  | 79.11 |
>
> 2) We **add experiments** on the text and image tokens on different **learning depth**. When we conducted ablation experiments on visual embeddings depth, the value of text embeddings depth remained at best value (9). The reverse is also the same.
>
> * Base-to-novel generalization of 11 datasets (Textual Tokens Depth)
> | Textual Tokens Depth      |  1     |  3     |  5       | 7      |  9     | 11    |
> |:---------------             |:------:|:------:|:--------:|:------:|:------:|:------:|
> | HM (Ours based on MaPLe)    |  77.87 |  80.00 |  78.90    | 79.02   | **80.32**  | 77.51 |
> | HM (Ours based on PromptSRC) |  78.15 |  79.90 | 80.16| 80.80   | **81.32**  | 78.23 |
>
>
> * Base-to-novel generalization of 11 datasets (Visual Tokens Depth)
> | Visual Tokens Depth      |  1     |  3      |  5       | 7      |  9     | 11    |
> |:---------------             |:------:|:------:|:--------:|:------:|:------:|:------:|
> | HM (Ours based on MaPLe)     |  76.06 |  77.03 |78.65 | 78.16   | **80.32**  | 76.93 |
> | HM (Ours based on PromptSRC)  |  76.91 |  78.22 |78.33 | 81.00   | **81.32**  | 78.09 |
>
> 3) If the reviewer has further suggestions on ablation experiments, we welcome discussions with reviewer.

---

> > ### Comment · Reviewer_xBkC · 2025-04-05
> >
> > Thank you for the author’s detailed response. I would like to revise my score upward.

---

> > > ### Author Response · Authors · 2025-04-09
> > >
> > > Dear Reviewer，
> > >
> > > Thank you for your time and effort in reviewing our paper. We are grateful for your feedback and pleased to hear your positive remarks!
> > >
> > > Best regards,
> > >
> > > Authors of #123

---

### Decision · Program_Chairs · 2025-05-01

**Decision:**

Accept (poster)

**Comment:**

This paper proposes an approach to improve generalization in target-unspecific tasks for CLIP-based models, leveraging Features Matrix (FM). The reviewers acknowledged the clear motivation, strong empirical results across a large number of datasets, plug-and-play compatibility, and thorough ablations. However, the more critical concern from the reviewers were about its unclear novelty over KgCoOp [Yao et al., 2023], which also regularizes tuning using pre-trained CLIP text features. While the proposed approach uses multiple prompts and contrastive loss on visual features, unlike KgCoOp which uses a single-prompt with L1 loss, the reviewers and the AC found the overall approach as conceptually very close and incremental. Another critical concern was its inference overhead from computing multiple cosine similarities, which may severely limit its practical applicability. Considering all these factors, I believe that this is a clear borderline paper, which could be accepted if there is enough room for presentation at the conference.